# Non-Small Cell Lung Cancer Treatment with Molecularly Targeted Therapy and Concurrent Radiotherapy—A Review

**DOI:** 10.3390/ijms24065858

**Published:** 2023-03-20

**Authors:** Katarzyna Król, Anna Mazur, Paulina Stachyra-Strawa, Ludmiła Grzybowska-Szatkowska

**Affiliations:** 1Department of Radiotherapy, St. John’s Cancer Centre, Regional Oncology Centre of Lublin, Jaczewskiego 7, 20-090 Lublin, Poland; 2Department of Radiotherapy, Medical University in Lublin, Chodźki 7, 20-093 Lublin, Poland

**Keywords:** EGFR, VEGFR, PD-1, PD-L1, NSCLC

## Abstract

Lung cancer is the leading cause of death worldwide for both men and women. Surgery can be offered as a radical treatment at stages I and II and selected cases of stage III (III A). Whereas at more advanced stages, combined modalities of treatment are applied: radiochemotherapy (IIIB) and molecularly targeted treatment (small molecule tyrosine kinase inhibitors, VEGF receptor inhibitors, monoclonal antibodies, and immunological treatment with monoclonal antibodies). Combination treatment, composed of radiotherapy and molecular therapy, is increasingly employed in locally advanced and metastatic lung cancer management. Recent studies have indicated a synergistic effect of such treatment and modification of immune response. The combination of immunotherapy and radiotherapy may result in the enhancement of the abscopal effect. Anti-angiogenic therapy, in combination with RT, is associated with high toxicity and should be not recommended. In this paper, the authors discuss the role of molecular treatment and the possibility of its concurrent use with radiotherapy in non-small cell lung cancer (NSCLC).

## 1. Introduction

Non-small cell lung cancer (NSCLC) is diagnosed at advanced stages (III or IV according to the American Joint Committee on Cancer (AJCC) staging system) of the disease in most of cases (about 75%), when radical treatment is often practically impossible [1]. According to the Polish Ministry of the Health data, the situation among Polish patients is even worse: their lung cancer treatment usually (in 45% to 62% of the cases, depending on the province) starts at the last stage of cancer development. The percentage of lung cancer cases diagnosed at stage I does not exceed 10% in any province of Poland [2]. A large group of patients with NSCLC are not eligible for radical surgery, not only due to the cancer disease stage at diagnosis but also because of their comorbidities implying a high risk of complications. The epidemiologic characteristics of NSCLC patients entail the need to search for more effective and more selective ways of systemic treatment and to optimize local and systemic treatment interactions. It is also essential to match the treatment equally to the histopathological and molecular type and stage of cancer and to the age, general condition, and comorbidities of the patient.

This review briefly presents molecular targets in NSCLC cells, examples of molecularly driven therapies used in NSCLC, and some reasons for the concurrent use of targeted therapies and radiotherapy in NSCLC. Then, results of some current clinical trials designed for the assessment of NSCLC concurrent treatment with targeted therapies and radiotherapy are provided.

## 2. Methodology

This review aimed at defining the current state of knowledge on the safety, effectiveness, and benefits of incorporating both targeted therapies and radiotherapy in the management of NSCLC. The specific objective was to analyze all clinical scenarios in which a combined treatment strategy was implemented and then to identify those bringing additional benefits compared to standard treatment. These benefits were assessed by analyzing parameters such as progression-free survival (PFS), overall survival (OS), and response rates (RRs). Having our questions defined—with reference to the participants, interventions, comparisons, outcomes, and study design—we could specify the source of information to be used in order to obtain answers. We decided to search for articles defined as trials, with the whole article (not only a summary) published, and only publications in English were included. We performed a query of the PubMed and PubMed Central databases using the following terms: ‘NSCLC’ (as the abbreviation and the whole name), ‘targeted therapy’ (plus ‘molecularly targeted therapy’), and ‘radiotherapy’ in conjunction mode. Then, we made an additional search using the following terms: ‘lung cancer’, ‘lung adenocarcinoma’, ‘adenocarcinoma of the lung’, ‘lung squamous cell carcinoma’, and ‘squamous cell carcinoma of the lung’.

Our additional source of information was the literature cited in the trials found in the database search. The choice of articles included in the review was based on some objective criteria, as mentioned above (only English trials, recruiting the NSCLC patients, intervention involving combined targeted therapy and radiotherapy, especially when compared to standard treatment, and results provided using a quantitative method), but also on some criteria of relevance to the subject, as defined by the authors, which we admit could have been a biased part of the selection.

## 3. Targeted Therapy

Molecularly targeted therapies applied to an appropriately selected population have been proven to be more effective and better tolerated than conventional chemotherapy. The OPTIMAL study, a Chinese phase III randomized clinical trial (RCT), was conducted in order to compare the effectiveness of erlotinib vs. gemcitabine-carboplatine chemotherapy in NSCLC patients with epithelial growth factor receptor (EGFR) mutation. The median PFS was found to be significantly longer for the erlotinib group compared to the group undergoing chemotherapy (13.1 months vs. 4.6 months; *p* < 0.0001). This observation was then confirmed in the EURTAC trial involving a European population [3,4].

These two clinical trials have established a new standard of management of disseminated adenocarcinoma (stage IV disease according to the AJCC classification) with activating mutation in EGFR gene: the EGFR thyrosine kinase inhibitor stands the first choice before chemotherapy in this group [5].

Molecularly targeted treatment is used in NSCLC in selected groups of patients with molecularly identified predictors of response to treatment. These drugs were summarized in Table 1.

## 4. Radiation Therapy

Radiation therapy is beneficial at every stage of NCSLC. Stereotactic ablative body radiotherapy (SABR) should be considered as an alternative to surgery at stage I of cancer, especially in fragile and medically inoperable patients [37,38]. SABR can be delivered safely due to the sophisticated methods of planning and delivering high doses of ionizing radiation to the precisely defined and carefully localized (or even tracked) volume of the tumor. A different range of dose is prescribed, depending on cancer tumor localization and its diameters, from 50–60 Gy in five fractions of 10–12 Gy in the centrally located tumors to 54 Gy in three fractions of 18 Gy in the tumors located peripherally in the chest [37,38]. Radiotherapy constitutes effective palliative treatment at advanced stages of the disease, in the alleviation of symptoms caused by the locoregional growth of neoplastic lesions and by distant spread of cancer. The total dose and fractionation depend on the predicted span of life, the number and localization of lesions, the patient’s performance status, and symptoms. Palliative radiotherapy of bone metastases with a single dose of 8 Gy has been proven to be equal compared with a scheme of 20 Gy in five fractions, as regards the pain control rate, but patients tend to be more often re-irradiated after a single dose treatment [39]. The current trial provided better results for SABR (24 Gy in two fractions) administered in painful spine metastases, compared to conventional irradiation (20 Gy in five fractions) [40]. Palliative radiotherapy administered to the chest is usually performed with 20 Gy in five fractions or 30 Gy in 10 fractions, but prolonged hypofractionation scheme is sometimes proposed, for example, 45 Gy in 15 fractions [41].

There is an emerging concept of oligometastatic or so-called low-burden cancer, with individualized therapeutic decisions, optimally made by a multidisciplinary team based on the patient’s life expectancy, performance, disease volume and localization of metastases, genetic alterations, and other predictive factors, as well as response to systemic therapy. The patients appropriately qualified for consolidative radiotherapy or SABR to all metastatic sites can benefit from the therapy not only in terms of local control but also in terms of OS prolongation [42,43]. Stereotactic radiosurgery (SRS) of brain metastases (15–24 Gy in one fraction) offers excellent local control without mental deterioration and is considered the first choice before whole brain radiotherapy (WBRT), especially in the era of magnetic resonance imaging and in patients treated with small molecules penetrating through the blood–brain barrier, such as crizotinib [44,45].

In stereotactic radiotherapy, thanks to the possibility of limiting the high dose to a small tumor volume, with a rapid dose decrease in the surrounding healthy tissues, high fractional doses are administered in a short period (e.g., 54 Gy in three fractions or 60 Gy in five fractions—fractionated stereotactic radiotherapy) [37,38] or in a single dose (15–24 Gy given once—stereotactic surgery) [45]. As a result of using a high fractional dose (i.e., >5 Gy) depositing more energy in the irradiated tissue, there is a greater amount of DNA damage, including double-strand DNA breaks (DSBs) that are the most difficult to repair. A greater amount of damage, occurring in a very short time, causes an increase in the percentage of cancer cells dying as a result of radiotherapy, mainly in the mechanism of apoptosis [46,47]. In addition, in radiotherapy with fractional doses of >20 Gy, the apoptosis of vascular endothelial cells occurs, which also causes tumor cell necrosis due to hypoxia [46,47,48]. The death of tumor cells as a result of DNA damage by ionizing radiation, especially in the mechanism of apoptosis, is immunogenic (ICD, immunogenic cell death) because it is associated with mechanisms inducing APC maturation (APC, antigen presenting cells), i.e., calreticulin exposure on the surface of tumor cells before apoptosis, ATP release during apoptosis, and the release of intranuclear HMGB-1 protein (HMGB1: high-mobility group box 1). Dendritic cells are recruited to dying tumor cells by ATP, engulfing tumor antigens when stimulated by calreticulin and presenting tumor cell antigens to T cells when stimulated by HMGB-1. Ultimately, APCs trigger the IFN-gamma-mediated, IL-17, and IL-1beta-dependent immune response whose effector cells—cytotoxic T cells—can destroy cancer cells, including those resistant to chemotherapy and radiotherapy [49,50]. The administration of the TLR-7 vaccine to infiltrating lymphoma, in combination with low-dose irradiation of this region (4 Gy in two fractions), has been found to result in the long-term remission of cancer lesions in some patients, indicating that low-dose radiation may also activate the immune response [51]. At the same time, one should not forget about the well-known immunosuppressive effect of radiotherapy, manifested by lymphopenia, often deep and long-lasting. The intensity of lymphopenia correlates with the irradiated area (extracranial radiotherapy, especially to the chest, with a larger irradiated volume) and with the total dose (radical radiotherapy with doses of >45 Gy compared to palliative radiotherapy, i.e., with usual doses below 36 Gy) [52,53,54].

Simultaneous chemotherapy is supposed to enhance the effect of radiotherapy through the so-called radiosensitization. The enhancement of radiotherapy results from the damage to the DNA helix and the stabilization of single strands of DNA, making them more sensitive to the effects of ionizing radiation [55]. Concurrent or sequential chemoradiotherapy is considered a standard management in locoregionally advanced NSCLC cases if no contraindications are present [56]. The total dose of 60 Gy in 30 fractions is prescribed to the primary tumor and metastatic mediastinal nodes, without elective irradiation of non-involved nodal stations [57]. Dose escalation has not been proven beneficial [58]. A higher RR (response rate) has been shown in concurrent (vs. sequential) chemoradiotherapy based on platine or taxane regimens, but at the expense of some serious side effects that are likely to occur [59]. Enhanced toxicity limits chemoradiotherapy application to some lung cancer patients because the majority of them suffer from many comorbidities such as respiratory failure, cardiovascular diseases, and age-related conditions. Current investigations are aimed at verifying the possibility of molecularly targeted therapy addition to radiotherapy being equally effective and safer compared to concurrent radiochemotherapy. A simultaneous administration of targeted therapy is aimed at intensifying apoptosis as well as strengthening the patient’s immune response. All these activities are undertaken to sensitize the cell to ionizing radiation and to prevent tumor repopulation, which is one of the reasons for treatment failure [46].

## 5. Radiobiological Bases of Combining Targeted Therapies with Radiotherapy in NSCLC Treatment

### 5.1. Anti-EGFR Treatment

Exposure of cells to ionizing radiation triggers the EGFR phosphorylation in the same way as the attachment of its ligands (EGF or tumor growth factor alpha (TGF-α)) to EGFR. This causes the activation of a number of signaling pathways responsible for enhanced cell proliferation, decreased apoptosis, stimulation of neo-angiogenesis, and DNA repair processes. These processes result in the phenomenon of increased repopulation during radiotherapy, an unfavorable factor, potentially contributing to treatment failure. Both in vitro and in vivo, the inhibition of EGFR (by anti-EGFR monoclonal antibodies or small-molecule EGFR kinase inhibitors) leads to significant radiosensitization. Patients with EGFR activating mutations may experience rapid tumor regression after the use of oral tyrosine kinase inhibitors and, therefore, due to reducing hypoxia, increased tumor cells’ radiosensitivity [60,61]. The most common so-called ‘classic’ activating mutations are deletions in exon 19 and a single L585R nucleotide swap in exon 21 of the tyrosine kinase domain region of the EFGR gene [61].

The study by Bonner et al. [62], which compared the strategy of adding cetuximab to radical radiotherapy with radiotherapy alone in the treatment of patients with advanced head and neck squamous cells cancer, established a new standard of care in the case of contraindications to simultaneous chemoradiotherapy present. A comparison of the median OS (49.0 months vs. 29.3 months, *p* = 0.03) and PFS (the hazard ratio, HR = 0.70, with probability, *p* = 0.006) has indicated a significant benefit of adding cetuximab to radiotherapy. In addition, the incidence of grade ≥3 complications, with the exception of acne-like rash and infusion reactions, did not differ significantly between the treatment groups, including mucosal reactions [62].

An impressive effectiveness of combining cetuximab with radiotherapy in the treatment of squamous cell carcinomas of the head and neck has prompted further research on the use of similar treatment strategies in other malignancies, including NSCLC.

### 5.2. Anti-Angiogenic Treatment

It has been proven that radiotherapy induces an increase in VEGF expression in neoplastic tumors. The increase in VEGF-C expression in lung cancer cells is probably due to the activation of the PI3K/AKT/mTOR (phosphoinositide 3-kinases/serine/threonine-specific protein kinases/mammalian target of rapamycin) signaling pathway [63]. VEGF stimulates the proliferation of vascular endothelial cells—neoangiogenesis—in which we deal with the disorganization of the vascular structure, which in turn contributes to hypoxia. Under hypoxic conditions, the indirect cytotoxic effect of radiotherapy is significantly reduced. Anti-angiogenic drugs may enhance the effectiveness of radiotherapy by limiting hypoxia [63]. In vitro studies of anti-angiogenic drugs in combination with RT have shown some improvement in the therapeutic index [63,64].

Models of glioblastoma multiforme (GBM) and melanoma have a chaotic network of microcirculatory vessels, which are known to be radiation resistant in vivo [65]. The addition of VEGFR signaling pathway inhibitor (e.g. ExFlk—soluble extracellular; Flk—a blocker of VEGFR-2) to radiation (3Gy) of GBM and melanoma tumor models has resulted in a detectable change in cancer cell phenotype. These cells exposed to the ExFlk and radiation had been reverted into the cells exhibiting a radiosensitive phenotype that is prone to apoptosis. The same effect has been observed among endothelial cells, even those decidedly radioresistant, such as human umbilical vein endothelial cells or human micro-endothelial cells. After combined exposure to radiation and a VEGFR inhibitor, endothelial cells showed an apoptosis-prone phenotype [65]. Damage to the tumor vasculature can lead to more than the additive ‘anti-tumor’ effect of combined treatment. At the same time, severe bleeding may occur, due to extensive injuries to the vessels. NSCLC tumor features associated with a high probability of clinically relevant hemorrhage in response to anti-VEGFR drugs were identified in 2004 as squamous cell histology, a lesion site near major vessels, necrosis, and tumor cavitation [66].

### 5.3. Immunotherapy

While radiotherapy is only local treatment, some distant effects of local radiotherapy (the regression of distant metastases, outside the irradiated area) have also been observed, and this has been called the abscopal effect. Radiotherapy and, in particular, the use of high fractional doses, as in the case of stereotactic radiotherapy, induce a generalized immune reaction. Preclinical studies indicate possible mechanisms for the occurrence of the abscopal effect, i.e., immunomodulation of both tumor cells and the microenvironment surrounding the tumor [67,68]. As a result of radiotherapy, there is both direct damage to the DNA strand and indirect damage through the generated free radicals. Direct DNA damage causes sections of double-stranded DNA in the cytoplasm, which stimulates cGAS (cyclic guanosine monophosphate-adenosine monophosphate synthase). cGAS binding to cytoplasmic DNA creates a second messenger cGAMP (cyclic guanosine monophosphate-adenosine monophosphate) which attaches to the STING protein (the adapter protein stimulator of interferon genes) [69,70]. This leads to the activation of a cascade of interferon (IFN) release: type I and then type II. IFN-I presents dendritic cells with antigens released from tumor cells to cytotoxic CD8+ (CD—clusters of differentiation) T cells. Activated T lymphocytes and NK (natural killers) cells cause the secretion of type II interferon, which increases the level of MHC II (major histocompatibility complex II) [70]. Thus, immunomodulation leads to an increased presentation of tumor antigens and the activation of immune response, especially cytotoxic T lymphocytes, which in turn may lead to the destruction of tumor cells in foci distant from the irradiated site. Higher fractional doses are supposed to cause a greater percentage of cells to be killed and thus release more tumor antigens. Lower doses of radiotherapy, however, reduce the expression of TGF-beta in the surrounding tumors outside the irradiated area, which allows the effector T cells and NK cells to perform their function [68,70].

Both the programmed death receptor (PD1) and its ligands (PDL1 and PDL2) are responsible for the inhibition of T cell proliferation and cytokine secretion. The interaction between tumor infiltrating lymphocytes and PD-L1 and PD-L2 ligands on cancer cells results in an increased expression of PD1. The consequence of these processes is the loss of cytotoxic functions by effector lymphocytes (so-called exhausted lymphocytes) [71,72]. Such ‘exhausted’ lymphocytes lose their ability to kill cancer cells or viruses. Another negative regulator of activated T lymphocytes is the molecule CTLA-4 (cytotoxic T lymphocyte associated antigen). CTLA-4 inhibits the appearance of the CD28 molecule stimulating a specific immune response on T lymphocytes, reducing the proliferation of T lymphocytes [71,73].

The combination of radiotherapy with anti-PD-1 or anti-CTLA-4 drugs aims to abolish the inhibitory effect of these molecules on the immune response. In recent years, there have been more reports of the regression of distant lesions after radiotherapy of the primary lesion or part of the metastatic lesions, and this seems to be related to the introduction of immunotherapy to the treatment of an increasing number of patients and the more frequent use of stereotactic body radiation therapy (SBRT) in oligometastatic disease/‘oligoprogression’ [70]. This indicates a possible interaction between radiotherapy and immunotherapy, but systematic observation and analysis of the results are necessary. Research is also underway on the optimal combination of both methods [67,68,69]. The effect of an increased immune response induced by radiotherapy can be eliminated by activating Three Prime Repair Exonuclease 1 (TREX1), which degrades damaged DNA and thus inhibits the formation of cGAS/STING [68,69]. The abscopal effect occurs in patients with a healthy immune system. Probably, the abscopal effect may be the result of complex reactions stimulating the immune system, and it seems that it is mainly due to the presentation of the MHC-II (major histocompatibility complex II) and tumor antigens by macrophages and the induction of a specific response of T lymphocytes (Th-helper and cytotoxic Tc) [68,69,70].

## 6. Overview of Clinical Trials Incorporating the Use of Monoclonal Antibodies or TKIs Targeting EGFR or VEGF in the Treatment of NSCLC

### 6.1. Palliative NSCLC Treatment

#### 6.1.1. Anti-EGFR Treatment

The results of studies of targeted therapies in combination with palliative radiotherapy was presented in Table 2.

In two phase I trials, Korean and Canadian, the safety of nimotuzumab (anti-EGFR monoclonal antibody), in combination with radiotherapy, was established [74,75]. The Korean study involved 15 patients with stage IIB-IV NSCLC who were not eligible for radical radiotherapy and chemotherapy due to their age or comorbidities [74]. Nimotuzumab was administered once a week for 8 weeks, simultaneously with palliative radiotherapy (30–36 Gy, 3.0 Gy fractionation). Maintenance treatment with nimotuzumab was continued until disease progression or unacceptable complications. The objective RR to treatment was 46.7%, and local control within the irradiated field was 100% [74]. In a Canadian study involving 18 patients with stage III-IV NSCLC, response to treatment was achieved in 66% of the patients, and disease control in the irradiated area was achieved in 94% [75]. Treatment toxicity was acceptable in both studies.

The authors of a Taiwanese study [76] combined radiotherapy and EGFR-TKIs in patients with advanced NSCLC. Adding radiotherapy to the EGFR inhibitor TKIs at an early stage of the disease was supposed to reduce the occurrence of drug resistance and to prolong PFS. In 25 patients with NSCLC in stage IIIB or IV, who showed a response to the initial treatment with EGFR-TKIs, multi-target radiotherapy was added. Single metastatic lesions were treated with tomotherapy according to the hypofractionated scheme of 40–50 Gy in 16–20 fractions. The RR was 84%, and the median PFS was 16 months. Moreover, 3-year OS was achieved by 62.5% of the patients [76].

In another phase II study, the hypothesis that the simultaneous use of erlotinib with radiotherapy could overcome the phenomenon of radiation resistance was verified [77]. Forty patients with newly diagnosed (stage III-IV) or recurrent NSCLC received a 3-week therapy with erlotinib (150 mg/d). One week after the start of drug administration, palliative radiotherapy was added (30 Gy in 10 fractions). The median OS and PFS were 5.2 and 3.2 months, respectively. Therefore, the study did not prove the benefit of adding erlotinib to standard palliative radiotherapy in patients with advanced NSCLC [77]. Additionally, the lack of improvement in the quality of life may be due to the higher toxicity of the combination of erlotinib and chest radiotherapy. In the study by Zhuang et al. [78], out of 24 patients with inoperable NSCLC in stage IIIA-IV who underwent simultaneous radiotherapy with erlotinib, nine (37.5%) developed radiation pneumonitis grade ≥2, including two cases (8%) of grade ≥3. Three patients (12.5%) died of bilateral radiation pneumonia. Therefore, in patients treated concomitantly with erlotinib and radiotherapy, the high probability of radiation-induced lung injury should be taken into account [78]. Atmaca et al. [79] presented a case report of a patient treated with afatinib in the next line of treatment in combination with radiotherapy of the mediastinum and primary tumor in metastatic NSCLC. A partial response was obtained in both the irradiated area and metastatic lesions with good tolerance [79].

#### 6.1.2. Anti-Angiogenic Treatment

The BEVA2007 study [80,81], where the mPEBev vs. mPE (cisplatin, metronomic etoposide, bevacizumab vs. cisplatin, metronomic etoposide) regimen was administered, showed the activation of cytotoxic T cell responses and promotion of dendritic cell activation in the group receiving bevacizumab [80]. Pastina et al. [81] conducted a retrospective analysis of 69 patients who were treated with the mPEBev regimen in the BEVA 2007 study. Forty-five of them also underwent palliative radiotherapy to the area of at least one metastatic lesion. Statistical data analysis (Log-rank test) showed a longer median survival in the irradiated group (chemoimmunotherapy vs. chemoimmunotherapy and RT: 12.1 +/−2.5 months vs. 22.12 +/−4.3 months]). Longer survival was associated with an increase in the percentage of activated dendritic cells (DCs) and memory T cells [80,81].

The results of the analysis suggest that tumor irradiation may prolong the survival time, presumably by inducing an immunomodulatory effect, which is the basis for further studies in this field.

**Table 2 ijms-24-05858-t002:** Targeted therapies combined with radiotherapy in palliative setting—results of the studies.

Trial ID, Ref.	Recruitment Criteria, Number of Patients Included in the Study	Treatment’s Scheme	Results	Conclusion
Choi HJ et al., 2011 [74]	NSCLC at IIB–IV stage; older or fragile patients, 15 pts	Palliative RTh + nimotuzumab (concurrent and maintenance)	RR = 46.7% LC = 100%	100 mg/m^2^ dose well tolerated; doses > 200 mg/m^2^ cause pulmonary toxicity;
Bebb G et al., 2011 [75]	NSCLC at III–IV stage; palliative chest RT proposed, 18 pts	like above	RR = 66% LC = 94%	well tolerated treatment—adverse effects only of grade 1 and 2
Chang CC et al., 2011 [76]	NSCLC at IIIB–IV stage; patients responding to gefitynib or erlotinib, 25 pts	RTh + gefitynib or erlotynib continuation RTh 40–50 Gy/16–20 fx.;	RR (to the RTh) = 84% PFS = 16 mth 3yOS = 62.5%	early RTh concurrent with TKIs may prevent TKIs resistance
Swaminath et al., 2016 [77]	NSCLC at III–IV stage or recurrent,40 pts	erlotynib over 3 weeks + RTh since 2nd week (30 Gy/10 fx)	predicted QoL improvement not achieved (LCSS: actual = −12.5 U; predicted = 17.5 U); MS = 5.2 mth PFS = 3.2 mth	lack of clear benefit in terms of QoL
Zhuang et al., 2014 [78]	NSCLC at III–IV stage, 24 pts	erlotynib + chest RTh (palliative or radical setting)	ILD of grade ≥ 2 in 37.5% of the pts; grade 5 (death) in 12.5% of the pts	concurrent treatment with erlotinib and chest RTh may be associated with more frequent RILI (ILD) occurrence
Atmaca et al., 2014 [79]	NSCLC at IV stage, 1 patient—a case presentation	afatynib + palliative RTh delivered to primary and mediastinum		PR of irradiated and metastatic (not irradiated) lesions (!)
Martino et al., 2016 Pastina et al., 2017 [80,81]	NSCLC at IV stage, 69 pts	mPEBev (metronomical ChTh cisplatin, etopozid + bewacizumab) +/− palliative RTh of one or a few distant metastases	significant improvement of MS in RT group (MS = 22.1 vs. 12.1 mth) PFS- no difference	bewacizumab treatment probably led to synergistic effect with RTh; immune response triggered with activating DCs and Tc—abscopal effect ?

Legend: ChTh—chemotherapy; ChRTh—chemoradiotherapy; DCc—dendritic cells; fx.—fractions; ILD—interstitial lung disease; LC—local control; LCSS—Lung Cancer Symptoms Scale; LPFS—local progression-free survival; MS—median survival; mth—months; NSCLC—non-small cell lung cancer; OS—overall survival; PFS—progression-free survival; PR—partial response; pts—patients; QoL—quality of life; RILI—radiation induced lung injury; RR—response rate; RTh—radiation therapy; Tc—cytotoxic T lymphocytes; TKIs—tyrosine kinases inhibitors.

### 6.2. Treatment of NSCLC with Radical Intent

#### 6.2.1. Anti-EGFR Therapy

The SCRATCH study [80] was designed to assess the toxicity of concurrent radiotherapy with cetuximab in inoperable NSCLC. The patients received induction chemotherapy (platinum-based) followed by intravenous cetuximab on a weekly basis with concurrent conventional fractionated radical radiotherapy [82]. Three of the 12 patients did not receive the full regimen: one died of bronchopneumonia while on treatment, and two patients refused to continue treatment (one for grade 3 asthenia, the other for grade 2 skin reaction). No early grade 3–5 complications other than those listed above were observed. One patient experienced a late radiation reaction of interstitial pneumonia, requiring steroid use and periodic oxygen therapy [82]. In another study—N0422—concomitant treatment with cetuximab and radical radiotherapy was used in a group of 57 patients with locally advanced NSCLC, not eligible for radiochemotherapy [83]. The primary endpoint was the proportion of patients who survived at least 11 months [83]. Only grade 3 and grade 4 complications were reported in 31 patients (fatigue, wasting, dyspnea, rash, dysphagia) [83]. A higher-than-expected percentage of patients (70%) surviving for 11 months, with an acceptable level of toxicity, allows the consideration of this treatment regimen as promising and indicates the direction of further research in the group of elderly patients and those with a worse performance status [83].

NEAR was another study evaluating the combination of radiotherapy with cetuximab in a group of patients with stage III NSCLC, not eligible for radiochemotherapy [84]. The treatment regimen included radical radiotherapy with the IMRT (intensity modulated radiotherapy) technique (56 Gy to elective nodal areas, 66 Gy to the tumor and affected lymph nodes) and cetuximab administered concomitantly with radiotherapy. After the completion of radiotherapy, cetuximab was administered as maintenance treatment for 13 weeks [84]. Each of the 30 patients included in the study [median age: 71 years] had at least one serious comorbidity, such as COPD (Chronic Obturative Pulmonary Disease) or coronary artery disease. Partial remission rates were reported in 19 of 30 patients (63%). The median PFS was 8.5 months (and median local PFS 20.5 months). The median OS was 19.5 months, and 1- and 2-year survival rates reached 66.7% and 34.9%, respectively [84].

According to CTCAE (Common Terminology Criteria for Adverse Events), the grade 3 or higher complications occurred in 36.7% of the patients and, with the exceptions of interstitial pneumonia (one patient, i.e., 3.3%), lobe pneumonia (three patients, i.e., 10%), esophageal inflammation (three patients, i.e., 3%), COPD exacerbations (3.3%), and pericardial effusion (3.3%), were most likely not related to treatment [84].

Simultaneous radioimmunotherapy (cetuximab) in patients with locally advanced NSCLC with unfavorable prognosis due to the performance status or comorbidities (Zubrod performance status 2, respiratory failure or comorbidities disqualifying from radiochemotherapy) was also evaluated in the SWOG S0429 study. The median OS was 14 months; the median PFS was 8 months; and RR was 47%. The study confirmed quite good tolerance of simultaneous radioimmunotherapy with cetuximab. The study found no correlation between treatment outcomes and higher levels of EGFR expression [85].

Researchers from the SLCSG (Swedish Lung Cancer Study Group) recruited 75 patients with stage III NSCLC for simultaneous cetuximab and radical radiotherapy, preceded by induction chemotherapy [86]. The patients received two courses of chemotherapy (docetaxel and cisplatin) followed by cetuximab for 7 weeks, every 7 days, during which they underwent radical radiotherapy (up to a dose of 68 Gy). The study had low toxicity compared to most concomitant radiochemotherapy regimens; the median survival was 17 months, and 1-, 2-, and 3-year OS rates were 66%, 37%, and 29%, respectively [86].

In a study by Martinez et al. [87], the efficacy and safety of adding erlotinib to conformal 3D (three dimension) radiotherapy was tested in a group of 60 patients, assessing selected parameters (complication rate, Cancer Specific Survival (CSS), Complete Response (CR), Overall Response Rate (ORR), PFS, and OS) in comparison with the control group (30 patients treated with radiotherapy alone). The study group achieved higher CSS and CR indices compared to the control group. However, there were no differences in OS, PFS, and ORR between the two groups. The rate of complications was significantly higher in the group of patients undergoing combined treatment. The addition of erlotinib to radiotherapy resulted in only insignificant clinical benefits, with a significant increase in toxicity. It seems that further studies on the use of erlotinib with radiotherapy should not be continued without prior identification, by biomarker analysis, of patients who could benefit from the EGFR-TKI therapy [87].

In a study by Lilenbaum et al. [88], the tolerability and effectiveness of sequential radiochemotherapy combined with erlotinib were assessed in locally advanced NSCLC in a group of patients with a poor performance status (usually PS 2) and/or significant weight loss. The RR was 67%, and disease control was achieved in 93%. Induction chemotherapy (carboplatinum and nab-paclitaxel) was administered, followed by radiotherapy combined with erlotinib. The treatment was well tolerated. The results achieved are higher than expected in this group but do not meet the criteria of statistical significance [88].

The use of EGFR-TKIs is the optimal method of treatment in patients with advanced NSCLC with the presence of an activating mutation in the EGFR gene. Unfortunately, in most cases, cancer cells acquire resistance to TKIs and thus to the disease progression. In a study by Wang et al. [89,90], radiotherapy was used, and treatment with a tyrosine kinase inhibitor was continued despite local progression after TKI applied as the first line of treatment. A total of 50 NSCLC foci in 44 patients were irradiated. The RRs and local control rates were 54.0 and 84%, respectively. The median OS was 26.6 months. Simultaneous conformal radiotherapy of measurable lesions in the chest, combined with EGFR-TKI, seems to be a reasonable and effective option in patients with advanced EGFR-mutated NSCLC, even after local failure of TKI in primary therapy [89,90].

#### 6.2.2. Anti-Angiogenic Treatment

In a study by Lind et al. [91], increased pulmonary toxicity of bevacizumab used simultaneously with radiotherapy was observed, in the form of a high percentage of radiation pneumonia. Before simultaneous radiotherapy with bevacizumab, patients received two cycles of platinum-based chemotherapy. The risk of pulmonary toxicity of radiotherapy was limited by accepting only plans meeting the V20 criterion (lung volume receiving a dose of 20 Gy or higher) of 36.8%. Only six patients were included in the study because four (67%) of them developed radiation pneumonitis grade 2 or 3 according to CTCAE [91]. In patients at stage III of this disease treated with radical radiochemotherapy, the rate of radiation pneumonia of grade 2 or 3 is 25% and 16%, respectively, and it significantly increases in patients with high V20 indices for the lungs. Apart from the small representation of the study group, the occurrence of grade 2 and 3 pulmonary toxicity in 67% is alarming, especially since there was no concurrent chemical treatment and since the V20 index did not exceed the acceptable value. The pulmonary toxicity observed was characterized by changes limited to the irradiated volume, with cavity formation in two out of four patients. The use of radiotherapy in combination with bevacizumab raises significant safety concerns [91].

The results of the studies of targeted therapies used in combination with radical radiotherapy were presented in Table 3.

### 6.3. NSCLC Treatment Escalation—Targeted Therapies and Radiochemotherapy Used Together

#### 6.3.1. Anti-EGFR Treatment

The results of targeted therapies combined with radiochemotherapy in the radical treatment intensification attempts were presented in Table 4.

The RTOG 0324 study was addressed to a group of patients with a good performance status, without significant weight loss, and with normal respiratory and organ capacity [95]. In this group, an intensive regimen was tested: simultaneous immunochemoradiotherapy followed by maintenance immunochemotherapy up to 17 weeks of treatment. According to the protocol, 80% of the patients received cetuximab together with chemoradiotherapy, and 86% received radiotherapy as planned. The RR to treatment was 62%; the median survival was 22.7 months; and the 2-year OS was 49.3% (survival longer than 41% as previously reported by RTOG (the Radiation Therapy Oncology Group)). There were five treatment-related deaths, and grade 3–4 CTCAE was presented (Table 4) [95].

The effect of intensification of the treatment regimen by adding cetuximab to concurrent radiochemotherapy in patients with stage III NSCLC was assessed in a multicenter randomized phase III RTOG 0617 study [96]. The primary endpoint of the study was OS, and there was no effect of adding cetuximab to chemoradiotherapy on that parameter (the median OS was 25 months in the cetuximab group vs. 24 months in the group without cetuximab). The hazard ratio (HR) equaled 1.07 (*p* = 0.29). The addition of cetuximab resulted in an increase in the rate of grade ≥3 complications (86% vs. 70%; *p* < 0.0001). There were also more deaths in the cetuximab group (10 vs. five patients). The second studied factor intensifying treatment; i.e., escalation of the dose of radiotherapy (to 74 Gy) did not bring the expected benefit either and even turned out to be potentially dangerous [96].

In the RTOG 0839 study (a phase II study involving 71 patients with locally advanced, potentially operable, stage III N2 NSCLC), the efficacy and toxicity of intensifying the trimodal treatment regimen by adding panitumumab were assessed [97]. In both groups of the RTOG 0839 study, patients received chemotherapy (carboplatin and paclitaxel) simultaneously with radiotherapy (up to a dose of 60 Gy, conventional fractionation), surgery (in the absence of progression and complications disqualifying from surgery), and two courses of adjuvant chemotherapy. In the study group, panitumumab was administered at a dose of 2.5 mg/m^2^ for 6 weeks during radiochemotherapy. Primarily, the rate of complete histopathological responses in the mediastinal lymph nodes (pCR—pathologically confirmed Complete Response) was assessed, which was 50% in the panitumumab-treated chemoradiotherapy group and 58.2% in the group not receiving panitumumab (*p* = 0.95) [97]. The study was discontinued during enrollment due to the lack of clear benefit and an unacceptable number of complications in the study group. Totals of 86% of the patients after induction chemoradiotherapy and 76% after immunochemoradiotherapy were operated on [97]. Postoperative grade 4 complications occurred in 13.6% of the patients in the group not receiving panitumumab and in 15.8% of those receiving the drug, while the occurrence of grade 5 complications (death) was 0% and 10.5%, respectively. It is not clear whether the high mortality is directly related to the use of panitumumab. There was also no increase in time to progression or the median OS [97]. Zaorsky et al. [98] drew attention to the case of a patient in whom a partial pathological response was observed with the elimination of a clone of cells with the G12D KRAS mutation. This is a potentially significant predictor of response to radiochemotherapy combined with panitumumab in NSCLC [98].

In another phase I study, also regarding the safety of gefitinib with radiochemotherapy, 16 patients with unresectable stage III NSCLC in a good general condition underwent combination treatment. Radiotherapy up to a dose of 70 Gy was performed simultaneously with gefitinib (250 mg/day, p.o.) and docetaxel (once a week). In addition, 2 weeks after the end of radiotherapy, two cycles of consolidation chemotherapy with docetaxel were administered, and gefitinib was continued until disease progression or the end of the study. The RR was 46%, and the median survival time was 21 months. [99]. This schedule of treatment appears to have moderate toxicity when the weekly docetaxel dose does not exceed 20 mg/m^2^ [99].

Further studies on the possibility of intensifying combined treatment of unresectable NSCLC with gefitinib include the study by Stinchcombe et al. involving simultaneous radiochemotherapy based on carboplatin and paclitaxel combined with gefitinib, preceded by induction chemotherapy [100]. The main adverse events observed during the study concerned stage III. The partial RR was 24%, and disease stabilization was achieved in 76% of the cases. The median PFS and OS were 9 and 16 months, respectively. The studied treatment regimen can be described as fairly well tolerated, with an acceptable side effect profile. However, the results regarding the achieved survival time are disappointing [100].

The CALGB-30106 study evaluated the addition of gefitinib to sequential and simultaneous radiochemotherapy in patients with unresectable stage III NSCLC. Lung adenocarcinoma accounted for 30% of histopathological diagnoses [101]. A total of 63 patients included in the study received two cycles of induction chemotherapy with paclitaxel, in combination with gefitinib. Patients with poor prognosis factors (weight loss ≥ 5% and/or performance status assessed as Zubrod 2) were then treated with radiotherapy (66 Gy in 33 fractions of 2 Gy) with gefitinib [101]. A group of patients with better prognosis (weight loss < 5%, performance status 0–1) underwent simultaneous radiochemotherapy (paclitaxel and carboplatinum) with gefitinib. After a resolution of adverse events in grade ≥ 2, gefitinib was introduced as a consolidation treatment. In the group with a worse prognosis, the median PFS and OS were 13.4 and 19.0 months, respectively, while in the group of patients with a better prognosis, the median PFS was only 9.2 months, and the median OS was only 13 months. Thirteen of the 45 samples analyzed were found to have an activating EGFR mutation: the L858R mutation in exon 21 in seven patients and the deletion in exon 19 in six patients. In addition, two patients had the T790 M mutation, which confers resistance to gefitinib, and these patients were excluded from the group of patients with the presence of an activating mutation in the statistical analysis. KRAS mutation was detected in seven out of 45 samples. There was no apparent difference in survival time between patients with EGFR/KRAS mutations and wild-type gene variants [101]. The results achieved in the group of patients with unfavorable prognostic factors treated with sequential chemoradiotherapy with gefitinib seem promising. However, short survival times in the group with a better prognosis receiving simultaneous chemoradiation with gefitinib bring disappointment. This also applies to patients with EGFR activating mutations [101].

The aim of the phase I study by Choong et al. was to determine the maximum tolerated dose of erlotinib in the combined treatment of patients with unresectable stage III NSCLC [102]. Erlotinib was administered only during radiochemotherapy, in increasing doses from 50 to 150 mg/day. Patients were randomized to receive erlotinib in combination with cisplatin and etoposide plus radiotherapy (66 Gy, 2 Gy/d) followed by three cycles of docetaxel or to receive an induction chemotherapy with carboplatin and paclitaxel followed by chemotherapy (carboplatin and paclitaxel) in combination with erlotinib. In both groups, the escalation of the erlotinib dose to 150 mg/d turned out to be feasible. The predominant adverse event observed in both patient cohorts was grade 3 or 4 leucopenia. In addition, grade 3 toxicity was observed, including esophagitis, vomiting, ototoxicity, diarrhea, dehydration, and pneumonia. The median survival was 10.2 months for group A and 13.7 months for group B, respectively. Three-year OS was achieved by 53% of the patients with skin rash, compared to only 10% of the patients without rash [102]. The addition of erlotinib to radiochemotherapy did not result in an evident increase in the toxicity of combination therapy [102]. However, the short median OS indicates that further clinical trials should be conducted in the population with the presence of an activating EGFR mutation.

Komaki et al. [103] assessed the effectiveness of treatment intensification in patients with locally advanced, inoperable NSCLC by adding erlotinib to simultaneous chemoradiotherapy. Forty-eight previously untreated lung adenocarcinoma patients underwent radiotherapy (IMRT, 63 Gy in 35 fractions) with concomitant chemotherapy (paclitaxel, carboplatinum) and erlotinib. The regimen also included paclitaxel–carboplatin consolidation chemotherapy. The mean time to progression was 14 months, and no differences were found between the group of patients without and with the detected EGFR mutation. Treatment toxicity was acceptable. The median OS was 36.5 months, and 1-, 2-, and 5-year survival rates were 82.6%, 67.4%, and 35.9%, respectively, regardless of the EGFR status [103]. Although the results of the study in terms of toxicity and OS were promising, the PFS did not meet the estimated value.

In a retrospective study, Ramella et al. evaluated the efficacy and toxicity of radiochemotherapy with erlotinib in patients with locally advanced or metastatic NSCLC, previously treated with chemotherapy [104]. Esophagitis and grade 3–4 pulmonary toxicity occurred in 2% and 8% of the patients, respectively. The majority (65%) of the analyzed patients did not progress. The median OS and PFS were 23.3 and 4.7 months, respectively. A RR was achieved in 53.3% of the patients who had not previously responded to first-line chemotherapy, including 13.3% in complete remission [104].

#### 6.3.2. Anti-VEGF Treatment

Two phase II studies by Spigel et al. [105], conducted independently on non-small cell and small cell lung cancer, were designed to pre-evaluate the efficacy and safety of bevacizumab in combination with radiotherapy and chemotherapy. Both studies were discontinued due to increased toxicity. The development of tracheoesophageal fistulae and related mortality were observed in both groups [105].

Another study confirmed the unacceptable toxicity of combined treatment with bevacizumab in locally advanced, inoperable NSCLC, due to the possibility of bleeding. The investigators do not recommend adding bevacizumab to combination therapy [106].

Anti-angiogenic drugs, when used concurrently with radiotherapy, have been found to act synergistically [63,64,65,66]. The synergistic effect of combined treatment with ionizing radiation and anti-angiogenic molecules is not only enhancement of cancer cells response but also an increased probability of injury to the vessels and connective tissue surrounding the tumor as well.

**Table 4 ijms-24-05858-t004:** Targeted therapies combined with radiochemotherapy in the radical treatment intensification attempts—results of the studies.

Trial ID, Ref.	Recruitment Criteria, Number of Patients Recruited	Treatment’s Scheme	Results	Conclusion
Blumenschein et al., RTOG 0324, 2011 [95]	NSCLC at III stage; ECOG 0, without substantial weight loss and without significant comorbidities; 75 pts	cetuximab + ChRTh (63 Gy) → cetuximab + ChTh until 17 weeks of treatment	RR = 62%; MS = 22.7 mth; 2yOS = 49.3%; five deaths; grade 4 hematological adverse events—20% grade 3–4 pneumonitis—7% grade 3 esophagitis—8%	80% patients have been given whole the planned treatment which means good compliance; survival longer than reported by previous RTOG studies;
Bradley et al., RTOG 0617, 2015 [96]	NCSLC at III stage, 544 pts	cetuximab + ChRTh (60 vs. 70 Gy) vs. ChRTh (60 vs. 70 Gy)	MS 25 mth in ChRTh + cetuximab groups vs. 24 mth in ChRTh without antibody groups; Grade ≥ 3 AE: 86% ChRTh with cetuximab vs. 70% ChRTh alone	addition of cetuximab to the RchTh has increased level of toxicity, without survival rates improvement;
Edelman et al., RTOG 0839, 2017 [97]	NSCLC at IIIA stage (potentially resectable), 71 pts (of 94 pts planned, due to early recruitment closure)	ChRTh (60 Gy) ±panitumumab→ surgery → ChTh	PCR = 50.0% in panitumumab arm vs. 58.7% in ChRTh alone arm; death rate 10.5% in experimental arm vs. 0% in control arm;	early recruitment closure because of unacceptable toxicity and no improvement of treatment results
Rothschild et al., 2011 [94]	inoperable NSCLC at III stage, 9 pts	gefitynib + ChRTh (63 Gy, ChTh- cisplatin)	2 patients (22.2%) reactions limiting dose: (1) dyspnea, dehydration connected with neutropenia resulting in pneumonia (2) liver enzymes elevation;	significant level of toxicity has made treatment difficult to conduct as planned
Center et al., 2010 [99]	inoperable NSCLC at III stage, ECOG 0-1, 16 pts	gefitinib + ChRTh (70 Gy, docetaxel) → ChT+gefitinib	RR = 46%; MS = 21 mth; grade 3–4 AE: esophagitis—27%, pulmonary—20%	The scheme possible to deliver, moderate toxicity, docetaxel dose should not exceed 20 mg/m^2^/week
Stinchcombe et al., 2008 [100]	unresectable NSCLC at III stage, 23 pts	ChT (carboplatin, irinotecan, paclitaxel)→ gefitinib +ChRTh (74 Gy, carboplatin and paclitaxel)	PFS = 9 mth; MS = 16 mth; AE: grade 3 esophagitis– 19.5%, atrial fibrillation– 9.5%	quite well tolerated, but without improvement of survival and TTP;
Ready et al., CALGB 30106, 2010 [101]	unresectable NSCLC at III stage: ‘better prognosis’ group: ECOG 0-1, without substantial weight loss; ‘worse prognosis’ group: ECOG 2 or weight loss >5%, 63 pts	gefitinib + ChRTh (66 Gy, carboplatin and paclitaxel, concurrent or sequential depending on prognostic group) → gefitinib until progression	‘better prognosis’ group: PFS = 9,2 mth; MS = 13 mth; ‘worse prognosis’ group: PFS = 13,4 mth; MS = 19 mth; toxicity comparable to the literature data on ChRTh without TKIs	disappointing results in concurrent ChRTh group—no benefit achieved with gefitinib addition (even in the EGFR mutated group); promising results in ‘worse prognosis’ group, given sequential ChRTh with gefitinib;
Choong et al., 2008 [102]	unresectable NSCLC at III stage, 17 pts	erlotinib (50/100/150 mg) + ChRTh (66 Gy, arm A: cisplatin and navelbin or arm B: k carboplatin and paclitaxel)	MS = 10.2 mth arm A; MS = 13.7 mth arm B; 3y OS = 53% in a group experiencing rash, 10% in a group with no rash;	manageable even at a dose of 150 mg erlotinib, without noticeable increase in toxicity; survival rates disappointing, there is a premise to withdrawal from further testing such a scheme;
Komaki et al., 2015 [103]	locally advanced, inoperable NSCLC, 48 pts	erlotinib + ChRTh (63 Gy, carboplatin and paclitaxel)	MTTP = 14 mth MS = 36.5 mth 1yOS = 82.6% 2yOS = 67.4% 5yOS = 35.9% AE: grade 5: zero patients, grade 4: one patient, grade 3: 11 patients;	low toxicity and long overall survival, but primary endpoint of the study—MTTP—lower than expected
Ramella et al., 2013 [104]	locally advanced or disseminated NSCLC, previously ChTh treated, 60 pts	erlotinib + ChRTh (primary tumor RTh)	grade 3–4 AE: esophagitis– 2%, RILI–8%; MS = 23.3 mth; PFS = 4.7 mth; no activating mutation in the EGFR gene present (but only 32% of the patients tested)	manageable scheme, however, recruitment to further studies should be based on identification of population with activating mutations in the EGFR gene confirmed
Spigel et al., 2010 [105]	NSCLC, 5 patients (early recruitment closure because of safety regards)	ChRTh + bevacizumab	tracheoesophageal fistulae formation	high risk of life-threatening reactions
Wozniak AJ et al., SWOG S0533 2015 [106]	unresectable stage III NSCLC ECOG PS 0-1, 26 pts: 11 of ‘High Risk’ of bleeding and 15 of ‘Low Risk’ *	‘Cohort1’: ChRTh -> consolidation treatment with ChTh (DTX) and bevacizumab;‘Cohort 2’: ChRth+bevacizumab	grade 5 pulmonary hemorrhage: two patients (both of ‘High Risk’, one with squamous histology and second with cavitation of tumor), grade 3 gastrointestinal haemorrhage: one patient, grade 3 pneumonitis: two patients, grade 3 and 4 anemia: two patients;	Seven of ‘High Risk’ pts after completing ChRTh obtained consolidation treatment, and two of them died because of fatal hemorrhage—unacceptable toxicity. ‘Cohort 2’ has been limited to ‘Low Risk” pts, but the trial was closed due to slow accrual;

Legend: AE—adverse events; ChTh—chemotherapy; ChRTh—chemoradiotherapy; DCc—dendritic cells; DTX—docetaxel; fx—fractions; ILD—interstitial lung disease; LC—local control; LPFS—local progression-free survival; MS—median survival; mth—months; MTTP—median time to progression; NSCLC—non-small cell lung cancer; OS—overall survival; pCR—pathological complete response; PFS—progression-free survival; pts—patients; RR—response rate; RTh—radiation therapy; SCLC—small-cell lung cancer; Tc—cytotoxic T lymphocytes; TKIs—tyrosine kinases inhibitors; * ’High Risk’ of bleeding was defined as one of these characteristics: squamous histology, hemoptysis, tumor with cavitation and/or adjacent to a major vessel, ‘Low Risk’—absence of ‘High Risk’ features.

## 7. Combination of Radiotherapy with Immunotherapy/Immunochemotherapy

After years of a frustrating lack of significant progress in the treatment of non-small cell lung cancer, consolidation treatment with durvalumab (anti-PD-L1 drug) after radical radiochemotherapy has turned out to be a clear step forward. Both preliminary and later results of the PACIFIC trial have brought some hope for successful treatment of more than half of the patients with local-regional NSCLC. In the study group (concurrent radiochemotherapy and durvalumab maintenance treatment), the 2-year survival rate was 66.3%, compared to 55% in the control group (radiochemotherapy and placebo), and the PFS was 17.2 months compared to 5.6 months [107]. In the updated analysis, both the median OS and the median PFS were significantly longer in the durvalumab group after radiochemotherapy: the median OS of 47.5 vs. 29.1 months, respectively, and PFS of 16.9 vs. 5.6 months. The 5-year survival rate corresponded with the early observations, being 42.9% and 33.4% in the durvalumab and placebo groups, respectively; i.e., over 40% of the patients randomized to the durvalumab group survived 5 years, and 1/3 remained without signs of the disease progression [107]. Long-term survival in the group undergoing simultaneous radiochemotherapy alone was also significantly longer than in the study establishing radiochemotherapy as the standard of care in the group of patients who received radical radiochemotherapy, which may be due to advances in radiotherapy planning and implementation techniques [59,107,108].

The immunostimulating effect of radiotherapy and the occurrence of the abscopal effect became one of the reasons to combine this treatment with anti-PD1 drugs in the treatment of metastatic NSCLC [49,69,70,108,109]. The PEMBRO-RT phase III trial on metastatic NSCLC compared to treatment with pembrolizumab alone (anti-PD1 drug), with pembrolizumab administered after SRBT (3 x 8Gy). Both the median PFS and the median OS were more favorable for the combination therapy (mPFS = 6.6 months vs. 1.9 months, *p* = 0.19; mOS = 15.9 months vs. 7.6 months, *p* = 0.16) [109]. In a phase I/II randomized trial on metastatic NSCLC, pembrolizumab was administered alone or in combination with radiotherapy for metastatic lesions [110]. Stereotactic or conventional radiotherapy was used. The median PFS for pembrolizumab alone was 5.1 months compared to 9.1 months for the radiotherapy group (*p* = 0.52). In the analysis, depending on the radiotherapy regimen, in the group receiving pembrolizumab and SBRT, PFS was 20.8 months (95% CI, 17.7 to 23.9 months), and in the case of combination of pembrolizumab with conventional radiotherapy 6.8 months (95% CI, 3.0 to 10.7 months) [110]. The objective response rate (ORR) outside the irradiated field in the patients receiving pembrolizumab and SBRT was 38%, compared to 10% in the pembrolizumab and conventional RT group. Although the mechanisms leading to the formation of the abscopal effect are still unclear, it seems that they do not depend on the total dose of RT, but rather on the fractional dose. PFS was also influenced by the level of PD-L1 expression. The median PFS for low PD-L1 expression was 4.6 months for pembrolizumab and 20.8 months for pembrolizumab with RT (*p* = 0.004). However, with high expression and a lack of PD-L1 expression, PFS for pembrolizumab treatment was 20.6 and 14.2 months, respectively, and for combination treatment was 5.6 months (*p* = 0.49) and 7.8 months (*p* = 0.25) [110].

In a phase I study involving patients with metastatic NSCLC or malignant melanoma, pembrolizumab (anti-PD-1 antibody) was combined with two radiotherapy regimens (24 Gy × 3 vs. a single fraction of 17 Gy). The patients who progressed during treatment with pembrolizumab were also eligible for the study. The abscopal effect occurred regardless of the radiotherapy regimen used, even in patients with earlier progression on pembrolizumab [111]. The mechanism of the abscopal effect is not fully understood and seems to depend not only on the applied irradiation scheme.

## 8. Conclusions

Due to the significant discrepancies in the design of the studies, it is difficult to obtain a reliable analysis. The use of anti-EGFR monoclonal antibodies simultaneously with RT is an attractive option for patients with NSCLC who have contraindications to simultaneous chemoradiotherapy. Only a properly selected group of patients with a specific histopathological type and activating mutation in EGFR may benefit from treatment with EGFR tyrosine kinase inhibitors in combination with radiotherapy. Further clinical trials should also be more focused on molecular testing to isolate patients who benefit from this treatment with acceptable toxicity. The attempts to intensify treatment by adding an anti-EGFR antibody to radiochemotherapy did not bring the expected improvement in treatment results. Further clinical trials are needed to evaluate both the toxicity of this treatment and the therapeutic benefit. Anti-angiogenic therapy in combination with RT, according to the current state of knowledge, is associated with high toxicity, and this combination should be avoided. A promising treatment strategy is the combination of immunotherapy and radiotherapy, which may result in the enhancement of the abscopal effect, but the optimal timing and optimal sequence of the modalities of treatment are currently not defined.

At the current stage of research, it is difficult to indicate whether therapy directed against EGFR or immunotherapy may be more effective in combination with radiotherapy in the treatment of NSCLC. Some trials addressed this question with encouraging observations. Unfortunately, the groups of people surveyed were not big enough in such studies to compare the results with a statistical significance.

## Figures and Tables

**Table 1 ijms-24-05858-t001:** Molecular target therapy used in NSCLC.

Group of Drugs	Target	Specific Indications/Predictive Factors	Drugs Examples
small-molecule tyrosine kinase inhibitors	EGFR (human epithelial growth factor receptor) kinases [6]	patients with activating mutations in the EGFR gene	1st generation: erlotinib, gefitinib, 2nd generation: afatinib [7], dacomitinib [8], 3rd generation: osimertinib [9], rociletinib [10], lazertinib [11]
	ALK (anaplastic lymphoma kinase) [12]	patients with the ALK or ROS 1 gene rearrangement	1st generation: crizotinib [13], 2nd generation: ceritinib [14], alectinib [15], brigatinib [16], entrectinib [17], 3rd generation: lorlatinib [18]
	BRAF kinases (V-raf murine sarcoma viral oncogene homolog B1) [19]	in patients with the V600E mutation	vemurafenib [20], dabrafenib + trametinib [21]
	VEGFR kinases (vascular endothelial growth factor receptor kinases) [22]		vandetanib [23], sunitinib [24], sorafenib
	multiple kinases (mainly VEGFR)		nintedanib [25]
VEGF-Trap	VEGF-A and PlGF (*placental growth factor*)		aflibercept [26]
monoclonal antibodies	EGFR	patients with overexpression of the epidermal growth factor receptor on cancer cells	chimeric antibodies: cetuximab, humanized antibodies: nimotuzumab, matuzumab, human antibodies: panitumumab, necitumumab, zalutumumab [27]
	VEGF-A		bevacizumab [28]
	VEGFR		ramucirumab [29]
immunotherapy [30]	PD-1 (type 1 programmed death receptor)		nivolumab [31] and pembrolizumab [32]
	PD-L1 (PD-1 ligand)		atezolizumab [33], durvalumab [34,35], avelumab [36]
	CTLA-4 (cytotoxic T lymphocyte antigen 4)		tremelimumab [36], ipilimumab [31]

**Table 3 ijms-24-05858-t003:** Targeted therapies combined with radiotherapy in radical setting—results of the studies.

Trial ID, Ref.	Recruitment Criteria, Number of Patients Included in the Study;	Treatment’s Scheme	Resul	Conclusion
Hughes et al., SCRATCH, 2008 [82]	NSCLC at III stage, not amenable to resection, 12 pts	Induction ChTh → cetuximab + RTh (64 Gy)	grade 3–5 acute reactions in 2 out of 12 pts (grade 5 pneumonitis, grade 3 asthenia); grade 3 late reactions in 1 out of 12 pts (RILI)	scheme’s acceptable toxicity (according to the study’s authors)
Jatoi et al., N0422, 2010 [83]	locally advanced NSCLC, patients > 65 years old, ECOG ≥ 2, 57 pts	cetuximab + RTh (60 Gy)	MS = 15.1 mth; PFS = 7.2 mth; 31/57 patients grade 3 reactions	rate of patients alive for more than 11 months higher than expected (70% vs. predicted 50%)—promising scheme in a group of patients with contraindications to ChRTh
Jensen et al., NEAR, 2011 [84]	NSCLC at III stage, patients with significant comorbidities, 30 patients	cetuximab + RT (66 Gy) → cetuximab over 13 weeks	RR (PR) = 63%; 1y OS = 66.7%; 2y OS = 34.9%; MS = 19.5 mth; PFS = 8.5 mth; LPFS = 20.5 mth; 36.7% patients have reactions of ≥ 3rd grade	high RR and long MS and low toxicity– promising results in a group of patients with contraindications to ChRTh
Chen et al., SWOG S0429, 2013 [85]	locally advanced NSCLC, ECOG ≥ 2 respiratory failure or other significant comorbidities, 24 pts	cetuximab + RTh (64,8 Gy) → cetuximab over 2 years or until progression	RR = 47%; MS = 14 mth; PFS = 8 mth; 22.7% patients have non-hematological adverse events of grade ≥ 3	scheme tolerance quite good; 11 patients has been tested for level of expression of EGFR, there has been no correlation between level of EGFR expression and results of the treatment (!)
Hallqvist et al., SLCSG, 2011 [86]	NSCLC at III stage, 75 patients	Induction ChTh→ cetuximab + RTh (68 Gy)	MS = 17 mth; 1yOS = 66%; 2yOS = 37%; 3yOS = 29%; grade 3 esophagitis 1.3%; grade 3–5 pneumonitis 5.6%	low toxicity comparing to concurrent ChTh—promising results
Martinez et al., 2016 [87]	locally advanced, not amenable to resection NSCLC; contraindications to ChTh; not tested molecularly for activating mutation in the EGFR gene, 30 pts + 60 pts	erlotinib + RTh (60 patients) vs. Rth alone (30 patients)	No difference in OS, PFS, ORR; CR increase (21.4% vs. 41.5%); CSS increase, but non statistically significant (17.7 mth–RTh vs. 21.4 mth–RTh+ erlotinib); adverse events rate increase (mainly skin toxicity)	noticeably higher toxicity, without treatment results improvement; call for the EGFR mutation identification in order to denote those patients, for whom TKIs combined with Rth treatment could be beneficial;
Lilenbaum et al., CALGB 30605 (Alliance)/RTOG0972 (NRG), 2015 [88]	NSCLC at III stage, ECOG ≥ 2 or substantial loss of weight, 75 patients	Induction ChTh→ erlotinib + RTh	RR = 67% PFS = 11 mth MS = 17 mth 1yOS = 57% treatment well tolerated;	rate of patients alive for more than 12 months has been quite satisfactory, but predicted level of 65% has not been achieved, so there is no evidence for beneficial role of erlotinib concurrent with Rth provided;
Wang et al., 2011 [89]	NSCLC at III-IV stage, 26 pts	gefitinib/erlotinib + chest Rth (median dose 70 Gy, dose range 42–82 Gy)	LC = 96% PFS = 10.2 mth MS = 21.8 mth 1yOS = 57% 2yOS = 45% 3yOS = 30% grade 3 adverse events—20% (hematological, esophagitis, pneumonitis)	promising scheme because of acceptable toxicity profile
Wang et al., 2018 [90]	Locally advanced NSCLC with confirmed activating mutation in EGFR, locally progressing during EGFR—TKIs therapy, 44 pts	EGFR-TKIs continuation+ RTh (chest)	TTP = 21.7 mth vs. 16 mth PFS = 21.3 mth vs. 16 mth RR = 54% LCR = 84% MS = 26.6 mth no significant adverse events;	improvement of both TTP and PFS based on measurable lesions; probably RTh used concurrently with TKIs may prevent TKIs resistance
Lind et al., 2012 [91]	inoperable NSCLC at III stage, 6 pts (early recruitment closure because of safety regards)	Induction ChTh→ bevacizumab + RTh (66 Gy)	ILD (RILI, lung fibrosis) four consequently recruited patients—67% ! two of six patient-grade 3 pneumonitis, two of six patient-grade 3 pneumonitis,	very little study, remarkably high prevalence of pulmonary adverse events (67%) in cases of Rth without concurrent ChTh, comparing e.g., pneumonitis rate after concurrent chemoradiotherapy (16–25%) is a premise to avoidance of bevacizumab concurrent with chest Rth due to unacceptable toxicity
Deutsch E. et al., 2015 [92]	NSCLC at III stage, 56% adenocarcinoma histopathology, 26 pts	RTh + everolimus (week before RTh, during RTh and 3.5 weeks after RTh) →ChTh	PR = 41%; SD = 32%; 2y-OS = 31%; 2y-PFS = 12%; 5/26—ILD (1 fatal); escalation of a dose possible in spite of some complications—without relationship between adverse effects and dose prescribed: ILD, esophagitis;	everolimus dosage for further studies established at 50 mg once a week.; remarkable pulmonary toxicity
Niho et al., JCOG 0402, 2012 [93]	locally advanced, not amenable to resection NSCLC, 38 pts	Induction ChTh → gefitinib + RTh (60 Gy) → gefitinib maintenance;	RR = 73%; MS = 28.5 mth; 2yOS = 65.4%; 60.5% compliance to the trial scheme, without grade ≥2 ILD;	rate of patients treated accordingly to the planned scheme of the study has been lower than expected
Rothschild et al., 2011 [94]	NSCLC at III stage, not amenable to surgery, 5 pts	gefitinib + RTh (63 Gy)	grade 1–2 adverse events (skin, subcutaneous tissue) no pulmonary reactions;	well tolerated treatment
Zhuang et al., 2014 [78]	NSCLC at III-IV stage, 24 pts	erlotinib + chest RTh(45–66 Gy)	grade ≥ 2 ILD—37.5%; grade 5–12.5%	concurrent treatment with erlotinib and chest RTh may be associated with higher rate of ILD

Legend; ChTh—chemotherapy; ChRTh—chemoradiotherapy; CR—complete response; CSS—cancer specific survival; EGFR—epithelial growth factor receptor; ILD—interstitial lung disease; LC—local control; LPFS—local progression-free survival; MS—median survival; mth—months; NSCLC—non-small cell lung cancer; OS—overall survival; PFS—progression-free survival; PR—partial response; pts—patients; RILI—radiation induced lung injury; RR—response rate; RTh—radiation therapy; SD—stable disease; TKIs—tyrosine kinases inhibitors; TTP—time to progression: y-year.

## Data Availability

All data are included in the article.

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
