# Peer review of "Non-Small Cell Lung Cancer Treatment with Molecularly Targeted Therapy and Concurrent Radiotherapy—A Review"

_ijms, 2023, doi:10.3390/ijms24065858_

Round 1
Reviewer 1 Report
These authors need some figures and good paragraph leading sentence to help readers to save the readers’ time to catch the point faster. “On and on description of the literature” will make readers easy be very tired and thus, only such readers whose research topic is closely related the review area, may continue to read. Specific comments are listed below.
1. While in the tile and abstract, the molecularly targeted therapy is the main focus in this review article, while after a short Introduction, these authors put the Radiation therapy in NSCLC as the 1.1 and targeted treatment as 1.2 without first focusing on reviewing the molecularly targeted therapy? This is not consistent with the title and the abstract.
2. To define the abbreviation from the first time appearing the review article. However, I did not see this. For example, “Median PFS has”. This review does not see a defined PFS as progression-free survival” until one the lane of 272.
3. For 1.3 description, these authors should use either a Table or a figure to summary the data. In this way, it will save readers time and improving description format (but not writing like a table of content format).
4. Based on the current content in the Section 2, Why these authors need to make the organization as 2. I. and 2.1. Given that there was even not having a 2.2 section, it is unnecessary for such weird format. Similar problem for Section 5.
5. It is unusual: This reviewer saw “2. I.RADIOBIOLOGICAL”; “3. ANTIANGIOGENIC TREATMENT”; “4. IMMUNOTHERAPY”; “5. II, In clinical trials overview”; “6. TREATMENT OF NSCLC”; “7. TREATMENT ESCALATION”; “8. COMBINATION OF”; and “9. Conclusion”. Why use “I”, “II” and why some time all capital letters and sometime not.
6. Tables are not in a good format. For example, why not combine the first two columns into one as “Pts No, Trial ID, refs” to save a column for the last “Conclusion” column, which has much more words.
7. Section 6 is on and on, whether it is possible to divide them into 6.1, 6.2, …etc. (for example, based on the trial ID as one thought). Similar issue for Section 7 as well.
8. Check typo and English. For example, “Poland. [1].Many”; “IN NCSLC”; “stage of NCSLC”; “seems to appear”(this will give others no any confidence at all. Nobody uses these two words together); “2. I.RADIOBIOLOGICAL”; “. In a phase I study in patients”; etc.
Author Response
We thank the reviewer for his generous comments on the manuscript. We have edited the manuscript in accordance with the reviewers’ comments. The work was submitted for linguistic proofreading- cetificate has been atteched . We improved the manuscript by adding and removing some fragments.
Point-by-point responses to reviewer’s comments are listed below
Response to Reviewer 1
These authors need some figures and good paragraph leading sentence to help readers to save the readers’ time to catch the point faster. “On and on description of the literature” will make readers easy be very tired and thus, only such readers whose research topic is closely related the review area, may continue to read. Specific comments are listed below.
Ad. Chemotherapy doses deleted in order to shorten trials' description.
- While in the tile and abstract, the molecularly targeted therapy is the main focus in this review article, while after a short Introduction, these authors put the Radiation therapy in NSCLC as the 1.1 and targeted treatment as 1.2 without first focusing on reviewing the molecularly targeted therapy? This is not consistent with the title and the abstract.
Ad. 1. switch of paragraphs (1st molecular biology, 2nd radiotherapy)
- To define the abbreviation from the first time appearing the review article. However, I did not see this. For example, “Median PFS has”. This review does not see a defined PFS as progression-free survival” until one the lane of 272.
Ad .2. checking all the abbreviations for presence of full versions provided
- For 1.3 description, these authors should use either a Table or a figure to summary the data. In this way, it will save readers time and improving description format (but not writing like a table of content format).
Ad. 3. changing a mode of displaying data (about targeted therapies of different kind) into a table
- Based on the current content in the Section 2, Why these authors need to make the organization as 2. I. and 2.1. Given that there was even not having a 2.2 section, it is unnecessary for such weird format. Similar problem for Section 5.
Ad. 4. reordering numeration of paragraphs
- It is unusual: This reviewer saw “ I.RADIOBIOLOGICAL”; “3. ANTIANGIOGENIC TREATMENT”; “4. IMMUNOTHERAPY”; “5. II, In clinical trials overview”; “6. TREATMENT OF NSCLC”; “7. TREATMENT ESCALATION”; “8. COMBINATION OF”; and “9. Conclusion”. Why use “I”, “II” and why some time all capital letters and sometime not.
Ad. 5. change of unnecessarily 'all in capital letters' words
- Tables are not in a good format. For example, why not combine the first two columns into one as “Pts No, Trial ID, refs” to save a column for the last “Conclusion” column, which has much more words.
Ad. 6. A piece of information about the patients’ number has been moved to the column with 'recruitment rules', and thereafter the second column in all the tables was deleted in order to reduce their dimensions, according to your kind advice.
- Section 6 is on and on, whether it is possible to divide them into 6.1, 6.2, …etc. (for example, based on the trial ID as one thought). Similar issue for Section 7 as well.
Ad .7.
The section 6 title has been changed into: ‘Overview of clinical trials incorporating the use of monoclonal antibodies or TKIs targeting EGFR or VEGF in the treatment of NSCLC cases’, and then this section has been divided into three parts:
- 1- Palliative NSCLC treatment,
- 2 - Treatment of NSCLC with radical intent,
- 3. - NSCLC treatment escalation ‒ targeted therapies and radiochemotherapy used together.
As the next step, section 6.1 was also divided:
- 1.1. -Anti-EGFR treatment
- 1.2.- Anti-angiogenic treatment
Section 6.2 was divided in the same manner as above, that is into:
- 2.1. Anti-EGFR therapy
- 2.2 Anti-angiogenic treatment
All the sections have been shortened by the removal of some sentences.
- Check typo and English. For example, “Poland. [1].Many”; “IN NCSLC”; “stage of NCSLC”; “seems to appear”(this will give others no any confidence at all. Nobody uses these two words together); “ I.RADIOBIOLOGICAL”; “. In a phase I study in patients”; etc.
Ad.8. the English correction was made by the certified translation and edition service

Reviewer 2 Report
This review discusses the use of combining radiotherapy with targeted therapy in the treatment of lung cancer. The information presented is useful. However, there are many similar reviews in the literature. It is not clear what this article may add to the literature. It may be useful for the authors to identify one or two emerging strategies for in-depth discussions, including mechanisms of action and clinical successes. There may be more comparative analysis of different combination regimens, such as immunotherapy + RT vs targeted therapy + RT. The simple list of clinical trials does not provide a clear idea about the directions and promises of future treatment strategies. The abstract is very general and does not provide much substantive information about new treatment strategies of lung cancer.
Author Response
We thank the reviewer for his generous comments on the manuscript. We have edited the manuscript in accordance with the reviewers’ comments. The work was submitted for linguistic proofreading- cetificate has been atteched . We improved the manuscript by adding and removing some fragments.
Comments and Suggestions for Authors
Response to Reviewer 2
This review discusses the use of combining radiotherapy with targeted therapy in the treatment of lung cancer. The information presented is useful. However, there are many similar reviews in the literature. It is not clear what this article may add to the literature. It may be useful for the authors to identify one or two emerging strategies for in-depth discussions, including mechanisms of action and clinical successes. There may be more comparative analysis of different combination regimens, such as immunotherapy + RT vs targeted therapy + RT. The simple list of clinical trials does not provide a clear idea about the directions and promises of future treatment strategies. The abstract is very general and does not provide much substantive information about new treatment strategies of lung cancer.it would be interesting to add an insight into the doses used in clinical practice and any correlations between dose and physical, biological or molecular effects
The following sentences has been added to the abstract:
‘The combination of immunotherapy and radiotherapy may result in the enhancement of the abscopal effect. Anti-angiogenic therapy in combination with RT is associated with high toxicity and should be not recommended.’
In the ‘Radiotherapy’ section, some of molecular aspects of radiotherapy have been added - this has been highlighted in the text.
Due to quite diverse groups of patients in studies on simultaneous radiotherapy with immunotherapy and targeted therapy, it is extremely difficult to compare the obtained results of the studies. In conclusion, we added the following sentences:
‘Further clinical trials should also be more focused on molecular testing in order to isolate patients who benefit from this treatment with acceptable toxicity.
The attempts to intensify treatment by adding an anti-EGFR antibody to radiochemotherapy did not bring the expected improvement in treatment results. Further clinical trials are needed to evaluate both the toxicity of this treatment and the therapeutic benefit.
A promising treatment strategy is the combination of immunotherapy and radiotherapy, which may result in the enhancement of the abscopal effect, but the optimal timing and optimal sequence of the modalities of treatment is currently not defined.
At the current stage of research, it is difficult to indicate whether therapy directed against EGFR or immunotherapy may be more effective in combination with radiotherapy in the treatment of NSCLC. Some trials addressed this question with encouraging observations. Unfortunately, the groups of people surveyed were not big enough in such studies to compare the results with a statistic significance.’

Reviewer 3 Report
The paper is a very interesting work on the role of molecular treatment and the possibility of its concomitant use with radiotherapy in non-small cell lung cancer (NSCLC); however it needs an implementation.
Here are some more detailed comments:
- - The introductory part lacks bibliographic references (lines 22-26),
- Radiation Therapy In Ncslc (it would be interesting to add an insight into the doses used in clinical practice and any correlations between dose and physical, biological or molecular effects)
- Review Of Molecularly Targeted Therapies In Nsclc (I suggest you include references to help the reader delve into specific topics of interest)
- Being a review, it would be appropriate for the authors to describe the criteria used for the selection of the papers, the sources, the purpose and the methodology applied.
- Tab 1 and 2 "Trial ID, ref" occupies an unsuitable priority space, it would be advisable to move it to the last column on the right and reduce the dimensions in favor of an increase in the dimensions of other columns where there are scientific information useful to the reader.
- Tab 3 is not mentioned in the text and in any case occupies an unsuitable place in the text (at the end of the conclusions). It would be appropriate to quote it in the text and move it.
- Conclusions: it would be interesting to know the opinion of the authors, commenting the results of this review work with their respectable scientific experience.
- References: must be implemented
Author Response
We thank the reviewer for his generous comments on the manuscript. We have edited the manuscript in accordance with the reviewers’ comments. The work was submitted for linguistic proofreading- cetificate has been atteched . We improved the manuscript by adding and removing some fragments.
Point-by-point responses to reviewer’s comments are listed below
Response to Reviewer 3
Comments and Suggestions for Authors
The paper is a very interesting work on the role of molecular treatment and the possibility of its concomitant use with radiotherapy in non-small cell lung cancer (NSCLC); however, it needs an implementation.
Here are some more detailed comments:
- The introductory part lacks bibliographic references (lines 22-26),
Ad.1. the lacking bibliographic references has been added
- Radiation Therapy In Ncslc (it would be interesting to add an insight into the doses used in clinical practice and any correlations between dose and physical, biological or molecular effects)
Ad. 2. the implied part has been added into the manuscript, according to your kind suggestion; all the added phrases has been highlighted;
- Review Of Molecularly Targeted Therapies In Nsclc(I suggest you include references to help the reader delve into specific topics of interest)
Ad. 3.the bibliographic references to the section mentioned (now enlisted in the form of Table 1) has been included
- Being a review, it would be appropriate for the authors to describe the criteria used for the selection of the papers, the sources, the purpose and the methodology applied.
Ad. 4. the section ‘methodology’ has been added
- Tab 1 and 2 "Trial ID, ref" occupies an unsuitable priority space, it would be advisable to move it to the last column on the right and reduce the dimensions in favor of an increase in the dimensions of other columns where there are scientific information useful to the reader.
Ad. 5. A piece of information about the patients’ number has been moved to the column with 'recruitment rules', and thereafter the second column in all the tables was deleted in order to reduce their dimensions, according to your kind advice.
- Tab 3 is not mentioned in the text and in any case occupies an unsuitable place in the text (at the end of the conclusions). It would be appropriate to quote it in the text and move it.
Ad. 6. Table 3 has been quoted in the text and moved to the suitable place.
- Conclusions: it would be interesting to know the opinion of the authors, commenting the results of this review work with their respectable scientific experience.
Ad. 7. The following sentences were added:
A promising treatment strategy is the combination of immunotherapy and radiotherapy, which may result in the enhancement of the abscopal effect, but the optimal timing and optimal sequence of the modalities of treatment is currently not defined. At the current stage of research, it is difficult to indicate whether therapy directed against EGFR or immunotherapy may be more effective in combination with radiotherapy in the treatment of NSCLC. Some trials addressed this question with encouraging observations. Unfortunately, the groups of people surveyed were not big enough in such studies to compare the results with a statistic significance
- References: must be implemented
Ad. 8. the references has been implemented

Round 2
Reviewer 2 Report
This revised version is improved. It would be useful to provide some explanations or rationale for why combination of RT with anti-angiogenic therapy causes high toxicity in the main text, as this is a main point in the abstract. There is still editing needed, e.g., a comma is missing in the abstract.
Author Response
Response to Reviewer
We thank the reviewer for his generous comments on the manuscript. We have edited the manuscript according to the reviewer’s comments We improved the manuscript by adding some fragments. Point-by-point responses to the comments are listed below this letter.
Best regards Ludmiła Grzybowska-Szatkowska Katarzyna Król Anna Mazur Paulina Stachyra-Strawa
Response to Reviewer
This revised version is improved. It would be useful to provide some explanations or rationale for why combination of RT with anti-angiogenic therapy causes high toxicity in the main text, as this is a main point in the abstract. There is still editing needed, e.g., a comma is missing in the abstract.
The following sentences were added:
Section 5.2. Anti-angiogenic treatment
Models of glioblastoma multiforme (GBM) and melanoma have a chaotic network of microcirculatory vessels, which are known to be radiation resistant in vivo [65]. The addition of VEGFR signaling pathway inhibitor (e.g. ExFlk - soluble extracellular Flk - a blocker of VEGFR-2) to radiation (3Gy) of GBM and melanoma tumor models, has resulted in a detectable change of cancer cells phenotype. These cells exposed to the ExFlk and radiation had been reverted into the cells exhibiting a radiosensitive phenotype, that is prone to apoptosis. The same effect has been observed among endothelial cells, even those decidedly radioresistant, like human umbilical vein endothelial cells or human micro endothelial cells. After combined exposure to radiation and a VEGFR inhibitor, endothelial cells showed apoptosis-prone phenotype [65]. Damage to the tumor vasculature can lead to more than the additive 'anti-tumor' effect of combined treatment. At the same time, severe bleeding may occur, due to extensive injuries to the vessels. NSCLC tumour' features associated with a high probability of clinically relevant haemorrhage in response to anti-VEGFR drugs were identified in 2004 as those: squamous cell histology, lesion site near major vessels, necrosis and tumour cavitation [ 66].
Section 6.3. NSCLC treatment escalation ‒ targeted therapies and radiochemotherapy used together - was divided into two sections :
6.3.1 Anti-EGFR treatment and 6.3.2 Anti-VEGF treatment
In section 6.3.2( Anti VEGF treatment ) added :
Another study confirmed the unacceptable toxicity of combined treatment with bevacizumab in locally advanced, inoperable NSCLC, due to the possibility of bleeding. The investigators do not recommend adding bevacizumab to combination therapy [106].

Reviewer 3 Report
The revised and implemented paper can be considered suitable for publication.
Author Response
We thank the reviewer for his generous comments on the manuscript. These comments helped us to improved our manuscript
Best regards Ludmiła Grzybowska-Szatkowska
Katarzyna Król
Anna Mazur
Paulina Stachyra-Strawa